# Conformational flexibility of HIV-1 envelope glycoproteins modulates transmitted/founder sensitivity to broadly neutralizing antibodies

Durgadevi Parthasarathy [1,11], Karunakar Reddy Pothula[2,11], Sneha Ratnapriya[1], Héctor Cervera Benet[1], Ruth Parsons [2,3], Xiao Huang[2], Salam Sammour[2], Katarzyna Janowska[2], Miranda Harris[1], Joseph Sodroski [4,5], Priyamvada Acharya [2,3,6] & Alon Herschhorn [1,7,8,9,10] ✉

HIV-1 envelope glycoproteins (Envs) of most primary HIV-1 strains exist in closed conformation and infrequently sample open states, limiting access to internal epitopes. Thus, immunogen design aims to mimic the closed Env conformation as preferred target for eliciting broadly neutralizing antibodies (bnAbs). Here we identify incompletely closed Env conformations of 6 out of 13 transmitted/founder (T/F) strains that are sensitive to antibodies that recognize internal epitopes typically exposed on open Envs. A 3.6 Å cryo-electron microscopy structure of unliganded, incompletely closed T/F Envs (1059-SOSIP) reveals protomer motion that increased sampling of states with incompletely closed trimer apex. We reconstruct de novo the post-transmission evolutionary pathway of a second T/F. Evolved viruses exhibit increased Env resistance to cold, soluble CD4 and 19b, all of which correlate with closing of the adapted Env trimer. Lastly, we show that the ultra-broad N6 bnAb efficiently recognizes different Env conformations and exhibits improved antiviral breadth against VRC01-resistant Envs isolated during the first-in-humans antibody-mediated-prevention trial.

Interaction of HIV-1 envelope glycoproteins (Envs) with the cellular CD4 receptor and CCR5/CXCR4 coreceptor mediates virus entry into target cells[1–5]. HIV-1 Envs are expressed on the surface of HIV-1 virions as trimeric spikes, with each spike composed of three gp120 exterior glycoproteins non-covalently associated with three gp41 transmembrane glycoproteins[6]. HIV-1 Env trimers of most primary isolates prefer to adopt a metastable closed, pre-fusion conformation and transition, either spontaneously or in response to CD4 binding, to downstream conformations[7]. Transition to an open Env conformation is mediated by extensive structural rearrangements that result in: (a) outward displacement of the gp120 V1/V2 from the apex to the sides of the trimer; (b) exposure of V3 loop that becomes disordered but appears

[1]Division of Infectious Diseases and International Medicine, Department of Medicine, University of Minnesota, Minneapolis, MN, USA. [2]Duke Human Vaccine Institute, Duke University, Durham, NC, USA. [3]Department of Biochemistry, Duke University, Durham, NC, USA. [4]Department of Cancer Immunology and Virology, Dana-Farber Cancer Institute, Boston, MA, USA. [5]Department of Microbiology, Harvard Medical School, Boston, MA, USA. [6]Department of Surgery, Duke University, Durham, NC, USA. [7]Institute for Molecular Virology, University of Minnesota, University of Minnesota, Minneapolis, MN, USA. [8]Microbiology, Immunology, and Cancer Biology Graduate Program, University of Minnesota, Minneapolis, MN, USA. [9]The College of Veterinary Medicine Graduate Program, University of Minnesota, Minneapolis, MN, USA. [10]Molecular Pharmacology and Therapeutics Graduate Program, University of Minnesota, Minneapolis, MN, USA. [11]These authors contributed equally: Durgadevi Parthasarathy, Karunakar Reddy Pothula. ✉e-mail: aherschh@umn.edu

to stay at the trimer apex; (c) formation of a 4-stranded bridging sheet that connects the outer and inner domains of gp120, (d) exposure of the coreceptor-binding site (-bs)[8-15] and (e) structural changes at the gp120/gp41 interface that alter access to the gp41 fusion peptide[16,17]. HR1-specific ligands (i.e., C34 peptide) bind cell surface-expressed Envs after interaction with sCD4, suggesting that gp41 HR1 coiled coil is exposed on a CD4-bound Env trimer[18-20]. Subsequent engagement of the Env-CD4 complex with the CCR5 or CXCR4 coreceptor moves the Envs down the energy gradient on the entry pathway, culminating in the formation of a gp41 six-helix bundle that facilitates the fusion of viral and cellular membranes[21-24].

Functional, entry-compatible intermediates of HIV-1 Envs can be enriched by introducing amino acid changes in control residues (e.g., L193A) that are highly conserved across all clades and restrain Envs in a closed conformation in primary wild-type isolates[7,25]. Functional Env intermediates are associated with hypersensitivity to cold[7,18,25] and to some but not all ligands that recognize internal epitopes, which are exposed on open or partially open Env conformations (e.g., soluble CD4 (sCD4), antibodies such as 17b, 19b, E51, and T20 peptide). Thus, sensitivity of HIV-1 to different Env ligands can identify exposure of internal epitopes on the Env surface and serves to define different Env conformations (e.g., closed, intermediate, and open). Based on this concept, several studies have provided evidence that the Envs of some primary HIV-1 strains may preferentially adopt a more open conformation relative to other primary isolates[26-28]. For example, some antibodies directed against the V3 loop of gp120 (e.g., 19b) neutralize only HIV-1 that exhibit partially or completely open Env conformations, which expose the V3 loop[7]. Unexpectedly, these antibodies neutralize a subset of global, "difficult-to-neutralize" HIV-1 isolates as well as a minority of transmitted/founder (T/F) strains, which can establish in vivo HIV-1 infection[26,27]. Here we identify − incompletely closed Env

conformations −of some T/F Envs based on their sensitivity to internal-epitope antibodies, cold, and sCD4. Our highly sensitive approach assesses HIV-1 Env conformational flexibility and heterogeneity on native HIV-1 Envs on viral particles. We solve the cryo-EM structure of an unliganded, incompletely closed T/F (1059) SOSIP Env and identify structural determinants of protomer flexibility and increased sampling of incompletely closed states. To follow the fate of incompletely closed Envs in vivo, we reconstruct viruses at different time points during in-patient evolution based on consensus HIV-1 sequences for each time point. Lastly, we associate the ultra-broad and potent neutralizing activity of the CD4-binding site (CD4bs) bnAb N6 (ref. 29) with recognition of different Env conformations; we then evaluate the antiviral activity of N6 against VRC01-resistant strains with incompletely closed Envs isolated from the first-in-humans antibody-mediated prevention trial (HVTN 704).

## Results
### Incompletely closed T/F Envs
To define the contribution of different elements to opening of Env conformation, we generated 53 functional Env variants by introducing changes in control residues that restrain the closed Env conformation of HIV-1$_{JRFL}$. We specifically designed a panel of Env variants containing the L193A and I423A changes, which were identified in our previous studies[7,18,25], to enrich functional intermediates in this group. All Env intermediates were functional with an average infectivity of 243% in comparison with WT JRFL (Cf2Th-CD4/CCR5 target cells; range 3–2084%; Supplementary Table 1). We tested HIV-1 sensitivity to Env ligands and identified variants with >10-fold increase in sensitivity to ligands that recognize internal elements associated with opening of the Env trimer (sCD4, CD4-induced Ab 17b, V3-directed Ab 19b, V1/V2-directed Ab 902090, and gp41 HR1-directed T20 peptide; Fig. 1a and

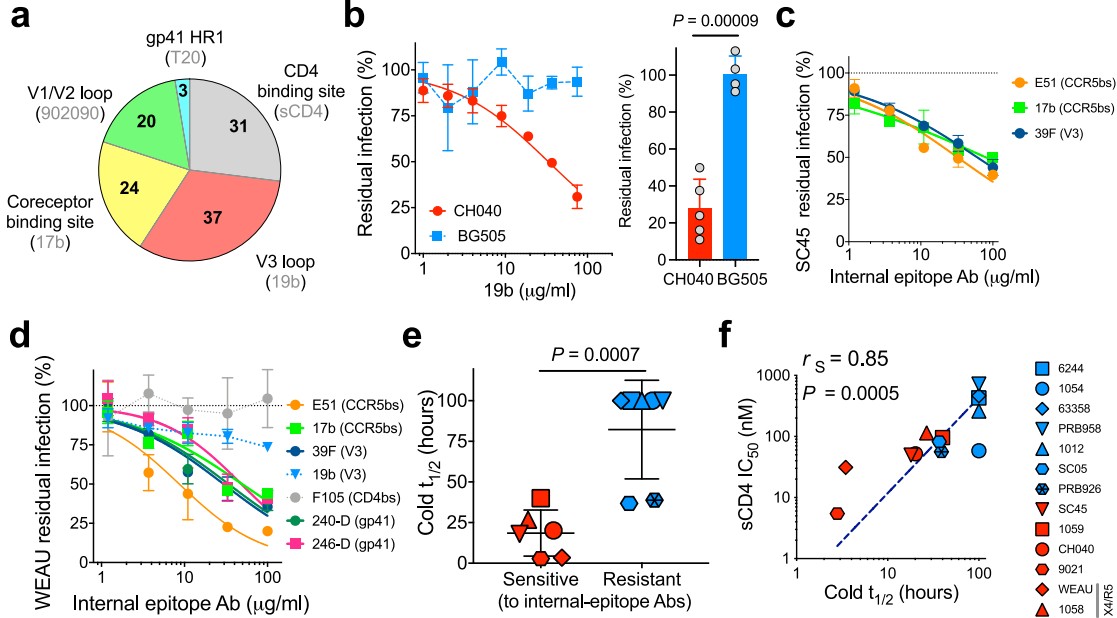

**Fig. 1 | Incompletely closed Env conformation of transmitted/founder (T/F) strains. a** We measured the sensitivity of WT and 53 selected HIV-1$_{JRFL}$ Env variants, which are enriched (not randomly selected) in intermediates, to ligands that recognize internal Env epitopes (sCD4; 17b, 19b, 902 antibodies; and T20 peptide) and calculated the contribution of each HIV-1 Env site exposure to trimer opening (number of variants hypersensitive to the specified antibodies; Supplementary Table 1 and Supplementary Fig. 1). Sensitivity of CH040 (**b**), SC45 (**c**), and WEAU (**d**) Envs to different antibodies recognizing internal epitopes. BG505 Envs were used as a control in (**b**). Data in (**b**–**d**) are mean values ± s.d.; *n* = 3 independent experiments each performed with 2–4 replicates (**b**, right), data are representatives of *n* = 3

(**b**, left) or 2 (**c**, **d**) independent experiments each performed with 2–4 replicates. Data in (**b** left) and (**b** right) are 2 independent sets of experiments. **e** Relationship between T/F Env sensitivity to internal-epitope antibodies and the sensitivity to cold. $t_{1/2}$ values higher than the maximal time tested (96 h) were set to an arbitrary value of 100 h. Relationship between cold and sCD4 sensitivities of T/F Envs that are sensitive (red) or resistant (blue) to internal-epitope antibodies. Dashed line is a log/log fit curve. $r_S$, Spearman correlation coefficient; Strain IDs in (**e**, **f**) are identical and specified in the symbol code on the right. *P*, two-tailed Student's *t* test *P* value (**b**, **e**) or two-tailed *P* value for calculated Spearman correlation coefficient (**f**). Source data are provided as a Source Data file.

Supplementary Table 1 and Supplementary Fig. 1). Analysis of the neutralization sensitivity of these 53 Env intermediates allowed us to map specific Env sites that are substantially exposed. We found that the gp120 V3 loop was the most frequently exposed element. In addition, among intermediates that exposed only a single Env element, those exposing only V3 were the most common (Supplementary Table 1), suggesting that in many cases the V3 loop may be the earliest element to be exposed during the opening of the Env trimer. Based on our results, we conclude that functional, entry-compatible HIV-1 Envs can expose single or different internal domains.

We next used these tools to analyze exposure of different sites on a panel of T/F Envs. HIV-1 T/Fs represent a subset of viral strains that can cross the mucosal bottleneck and establish HIV-1 infection in humans[27,30]. T/F virions display on their surface ~1.9-fold more Envs compared with chronic HIV-1 isolates and they are highly resistant to antiviral activity of α and β interferons[30,31]. We selected 13 previously published T/F Envs[27] (Supplementary Table 2), which exhibited high infectivity in vitro, and tested their sensitivity to ligands that recognize internal Env elements (E51, 17b, 39 F, 19b, 697-30D, F105, T20 peptide, 240-D, and 246D; Fig. 1b–d and Supplementary Fig. 2). We detected six sensitive T/F Envs that exposed either only the V3 loop based on 19b sensitivity (CH040, 1058 and 9021); the V3 loop and epitopes of the coreceptor-bs based on sensitivity to E51, 17b and 39 F (SC45 and 1059; 1059 was only weakly sensitive to 17b); or multiple internal epitopes (WEAU) (Fig.1b–d and Supplementary Fig. 2). Similar to our results with the 53 HIV-1$_{JRFL}$ variants, the V3 loop was the most frequently exposed Env element but T/F Envs were only weakly to moderately sensitive to internal-epitope antibodies, indicating that subset of Envs are incompletely closed and only slightly or temporarily expose internal epitopes. Consistent with an incompletely closed Env conformation, neutralization-sensitive viruses were significantly ($P = 0.0007$) more sensitive to cold exposure (Fig. 1e and Supplementary Fig. 3a, b) than the remaining T/F viruses in this group. More open Env conformations are more sensitive to inactivation in the cold[7,18,32]. T/F viruses with incompletely closed Envs were also more sensitive to sCD4, although this difference did not reach statistical significance ($P = 0.05$; Supplementary Fig. 3d); and we observed a significant correlation between cold and sCD4 sensitivity ($P = 0.0005$; Fig. 1f). Notably, two out of two dual-tropic (X4/R5) Envs (1058 and WEAU) in the T/F group were incompletely closed and hypersensitive to cold and sCD4. We next measured T/F virus sensitivity to bnAbs directed against known sites of Env vulnerability and detected diverse patterns of susceptibility (Supplementary Figs. 2, 4). We focused on 1059 Envs because viruses pseudotyped with these Envs were hypersensitive to bnAbs directed against the gp41 membrane-proximal external region (MPER), which preferentially neutralize more open Env conformations[7,33–35] and were highly resistant to VRC01, VRC03, 3BNC117, and PG9 (but not PGT145) bnAbs that exhibit significantly reduced activity against partially open, Env intermediates of primary HIV-1 strains[7,34] (Supplementary Table 3 and Supplementary Figs. 2, 4). To evaluate the integrity of bnAb epitopes in 1059 Envs we tested bnAb binding to soluble 1059 gp120, in which the CD4bs is fully accessible to antibodies. Despite resistance of 1059 Env trimers on virions to several CD4bs bnAbs, soluble 1059 gp120 strongly bound VRC01 and to a lesser extent 3BNC117, and this binding was comparable to the binding of these antibodies to gp120 of HIV-1$_{AD8}$, a strain which is hypersensitive to neutralization by CD4bs and V1V2 bnAbs (Supplementary Fig. 5a–d). In contrast, soluble 1059 gp120 only weakly-moderately bound VRC03. Thus, resistance of 1059 Envs to VRC01 and 3BNC117 is consistent with a conformational effect that limits target site accessibility in the context of the native Env trimer, whereas resistance to VRC03 may be related to a combined effect of epitope integrity and accessibility. Resistance of 1059 Envs to PG9 but not PG145 suggests a local effect. Both bnAbs target V1/V2 quaternary epitopes and therefore poorly bind soluble gp120. Nevertheless, binding of both bnAbs to 1059 gp120 was comparable to or

stronger than their binding to AD8 gp120 (Supplementary Fig. 5c). In comparison with a virus with tightly closed Envs (e.g., BG505), the incompletely closed 1059 Envs required lower concentrations of exogenous sCD4 to infect Cf2Th-CCR5 target cells (Supplementary Fig. 5e).

HIV-1$_{1059}$ Env resists VRC01 neutralization despite strong binding of soluble monomeric 1059 gp120 to VRC01 (Supplementary Fig. 5a–d), suggesting interference with access of VRC01 to its binding site on the native Envs on HIV-1$_{1059}$ virions. To gain insights into the structural features of the incompletely closed Envs identified in this screen, we determined cryo-EM structures of 1059-SOSIP, and for comparison, of the well-characterized BG505-SOSIP. Both the 1059- and BG505-SOSIP trimers were purified using *Galanthus nivalis* lectin (GNL) chromatography, which exploits lectin binding to Env surface glycans in a conformation-independent manner, followed by size-exclusion chromatography (SEC; Supplementary Figs. 6, 7). We obtained cryo-EM reconstructions of unliganded 1059-SOSIP, resolved at 3.6 Å with 737,588 particles, and of BG505-SOSIP resolved at 3.7 Å with 975,399 particles (Table 1). To assess variability in the datasets, we further classified each particle stack into 10 sub-classes. Analysis of the unliganded 1059-SOSIP sub-classes revealed three structural characteristics of the incompletely closed Envs (Fig. 2a–d): 1. conformational flexibility 2. asymmetry and 3. partial opening of trimer apex. Within these 10 subclasses, we observed a larger range of protomer motion in 1059-SOSIP compared with the limited range of motion observed in BG505-SOSIP (Fig. 2b and Supplementary Fig. 8). The 10 sub-classes obtained for 1059-SOSIP showed variable levels of asymmetry in the protomer arrangement that were significantly greater ($P$ value = 0.0019; Fig. 2c, Supplementary Fig. 8c,c' and Supplementary Tables 4, 5) than the asymmetry observed in the BG505-SOSIP sub-classes. Further, 3D variability analysis of 1059-SOSIP captured scissoring motion of the protomers, and partial opening of the trimer characterized by outward motion of V1/V2 loop from the trimer apex (Fig. 2d; Supplementary Movies 1a–f and 2a–f, PC1 and PC3). In comparison, and consistent with the lower range of protomer motion observed for BG505-SOSIP, the 3D variability analysis of BG505-SOSIP, revealed no significant opening motions of the SOSIP trimer. Overall, the 1059-SOSIP cryo-EM analysis reveals occupancy of a wider range of protomer conformations contributing to augmented outward motion of the V1/V2 loop at the trimer apex and higher asymmetry of the Env trimer, compared to BG505-SOSIP. Notably, both 1059-SOSIP and antibody PGT151 bound JRFL Env (PDB ID: 5FUU) structures are asymmetric, clustered together in a principal component analysis (Fig. 2e, f) and exhibited promoter scissoring. Taken together with our viral neutralization results, we conclude that the Envs of some T/F HIV-1 strains are incompletely closed.

## in vivo evolution of incompletely closed T/F Envs

We studied how a T/F virus with incompletely closed Env conformation evolved post transmission in an individual infected with HIV-1$_{CH040}$. We selected T/F CH040 Envs because (a) single genome-derived *env* sequences from the plasma were available for over >1700 days[27,36,37], (b) initial analysis showed that CH040 Envs were almost completely resistant to all tested bnAbs except for VRC01 and VRC03 while maintaining high infectivity (Supplementary Fig. 2), and (c) unexpectedly, entry mediated by CH040 Envs was sensitive to 19b, which targets the gp120 V3 loop, as well as to sCD4 and to cold exposure, all of which support an incompletely closed Env conformation (Fig. 3b, c and Supplementary Fig. 2). We analyzed 475 available *env* sequences (Supplementary Tables 6, 7), built consensus sequences, and reconstructed de novo pseudoviruses displaying Envs that represent 10 different time points during CH040 Env evolution in the infected individual. Reconstructed viruses exhibited a gradual increase in resistance to 19b, cold, and sCD4 over the course of infection (Fig. 3b, c and Supplementary Fig. 9), indicating that CH040 Envs were evolving

**Table 1 | Cryo-EM data collection, refinement and validation statistics**

| EMDB PDB | 1059-SOSIP 41246 8TGW | BG505-SOSIP 41244 8TGU |
|---|---|---|
| Data collection and processing | | |
| Magnification | x81,000 | x81,000 |
| Voltage (kV) | 300 | 300 |
| Electron exposure (e−/Å²) | 59.1 | 63.2 |
| Defocus range (µm) | 1.0–3.0 | 1.4–1.8 |
| Pixel size (Å) | 1.08 | 1.08 |
| Symmetry imposed | C1 | C1 |
| Initial particle images (no.) | 7,260,519 | 6,566,700 |
| Final particle images (no.) | 737,588 | 975,399 |
| Map resolution (Å) | 3.6 | 3.7 |
| FSC threshold | 0.143 | 0.143 |
| Map resolution range (Å) | 3–5 | 3–5 |
| Refinement | | |
| Initial model used (PDB code) | ab-initio | ab-initio |
| Model resolution (Å) | 3.6 | 3.7 |
| FSC threshold | 0.5 | 0.5 |
| Model resolution range (Å) | 3–5 | 3–5 |
| Map sharpening B factor (Å²) | −181.9 | −145.3 |
| Model composition | | |
| Non-hydrogen atoms | 13,449 | 14,880 |
| Protein residues | 1488 | 1719 |
| Ligands | 117 | 96 |
| B factors (Å²) | | |
| Protein | 48.61 | 21.67 |
| Ligand | 83.24 | 39.12 |
| R.m.s. deviations | | |
| Bond lengths (Å) | 0.005 | 0.004 |
| Bond angles (°) | 0.789 | 0.703 |
| Validation | | |
| MolProbity score | 1.14 | 1.33 |
| Clashscore | 1.61 | 1.84 |
| Poor rotamers (%) | 0 | 0 |
| Ramachandran plot | | |
| Favored (%) | 96.44 | 94.26 |
| Allowed (%) | 3.49 | 5.68 |
| Disallowed (%) | 0.07 | 0.06 |

to become more closed. We detected several changes in the V3 loop during evolution but the 19b epitope in CH040 and all 41 sequenced viruses that evolved in CH040 patient by day 1737 was intact (IxxxxGxxFYxR; x = any amino acid; Supplementary Table 7). The observed evolution pattern is consistent with ability of antibodies developed in *humans* infected with HIV-1 to exert a selection pressure to conceal exposed Env epitopes, as well as with previous studies that monitored SHIV evolution in rhesus macaques[38,39]. It took more than 1000 days and 25 amino acid changes (Supplementary Table 6) to evolve to a more closed phenotype, which is consistent with previous observations of the development of antibodies with modest neutralization breadth in this individual[37].

With this information, we next studied how the evolution pattern from incompletely closed Env to a more closed state affected virus sensitivity to bnAbs (Fig. 3d). CH040 Envs were completely resistant to the V1/V2 loop bnAbs PG9 and PGT145 (up to 20 µg/ml) and this

pattern was mostly unchanged over the course of viral evolution. Sequence analysis of CH040 Envs identified the amino acid Lys at position 160 instead of the conserved Asn, which is typically glycosylated and significantly contributes to V1/V2 bnAb sensitivity. Introduction of the Asn at position 160 rendered CH040 Envs highly sensitive to both PG9 and PGT145 (Supplementary Fig. 10). Interestingly, Lys at position 160 was not fixed during HIV-1 evolution in the infected individual but was replaced by an Arg rather than Asn during the course of evolution. Sensitivity of CH040 Envs to CD4bs VRC01 and VRC03 bnAbs was maintained over the time of infection, although the evolved virus at late time points was slightly more resistant than the T/F CH040 (Fig. 3d). The third CD4bs bnAb, 3BNC117, was ineffective against CH040 but could readily block infection of the evolved viruses. Similarly, CH040 Envs were completely resistant to the V3-glycan bnAbs but became hypersensitive to these antibodies during in vivo evolution. We identified the E332N change in evolved viruses that typically contributes to V3-glycan bnAb sensitivity[40]. However, additional changes were required to confer sensitivity, which may include the N334S to generate the N332 related glycosylation motif Asn-X-Ser/Thr, since introducing a single E332N change into CH040 Env resulted in a virus that was still resistant to PGT121 bnAb (Supplementray Fig. 10). Sensitivity of CH040 Envs to PGT151 was maintained for at least 1737 days whereas CH040 Env sensitivity to the gp41 MPER bnAbs varied; resistance to 4E10 was maintained throughout the course of evolution but the evolved viruses became more sensitive to 7H6 bnAb (Fig. 3d). Thus, in vivo evolution of the CH040 Envs resulted in a more closed conformation that was recognized better by a subset of bnAbs.

**CD4bs bnAb neutralization of different Env conformations**
Identification of incompletely closed T/F Envs led us to hypothesize that bnAbs that can recognize multiple Env conformations exhibit broader and more potent neutralization activity. We focused on CD4bs bnAbs that typically neutralize primary viruses (e.g., HIV-1_JRFL) with closed Env conformation and are less effective against engineered, functional Env intermediates of these strains that expose most internal Env epitopes associated with Env opening[25,34]. We studied how altering Env conformation affects viral sensitivity to five different CD4bs bnAbs (N6, 3BNC117, NIH45-46 G54W, VRC01, and VRC03) by calculating the change in neutralization efficiency (fold change of IC₅₀ values) of Env intermediates compared to the WT Env (Fig. 4a–d). As expected, all CD4bs bnAbs efficiently neutralized the WT HIV-1_JRFL as well as the lab adapted HIV-1_SF162. However, their ability to recognize conformational changes associated with HIV-1_JRFL Env intermediates significantly varied. N6 was the most tolerant to conformational changes exhibiting a potent and wide inhibition of the closed and functional intermediates with mostly minor differences between IC₅₀ values of these variants and the WT HIV-1_JRFL Envs (Fig. 4a–d). Of note, N6 bnAb was identified by screening B cells without prior selection for binding to soluble Envs and it exhibits an ultrabroad and potent inhibition profile[29]. Consistent with an exceptional breadth, N6 efficiently neutralized diverse viruses with WT (closed) and I423A (open-intermediate) Env conformations from clades A, B, C and D (Fig. 4e). In contrast, VRC01, VRC03, and 3BNC117 bnAbs preferentially neutralized primary viruses with closed Env conformations and exhibited a significantly reduced inhibition activity against most HIV-1_JRFL Env intermediates (Fig. 4c, d)[7,25]. To assess this phenotype in a broader context, we analyzed hundreds of HIV-1 strains in HIV database[41] and studied the relationship between CD4bs bnAb breadth (% strains inhibited with IC₅₀ < 50 µg/ml), potency (expressed as geometric IC₅₀) and conformational flexibility. We detected a positive correlation between CD4bs bnAb breadth and potency, and a statistically significant correlation ($P < 0.05$) between CD4bs bnAb breadth and their ability to neutralize Env intermediates (i.e., J-I423A; Fig. 4f). This analysis provides a potential link between exceptional breadth of N6 (ref. 29) and neutralization of multiple Env conformations.

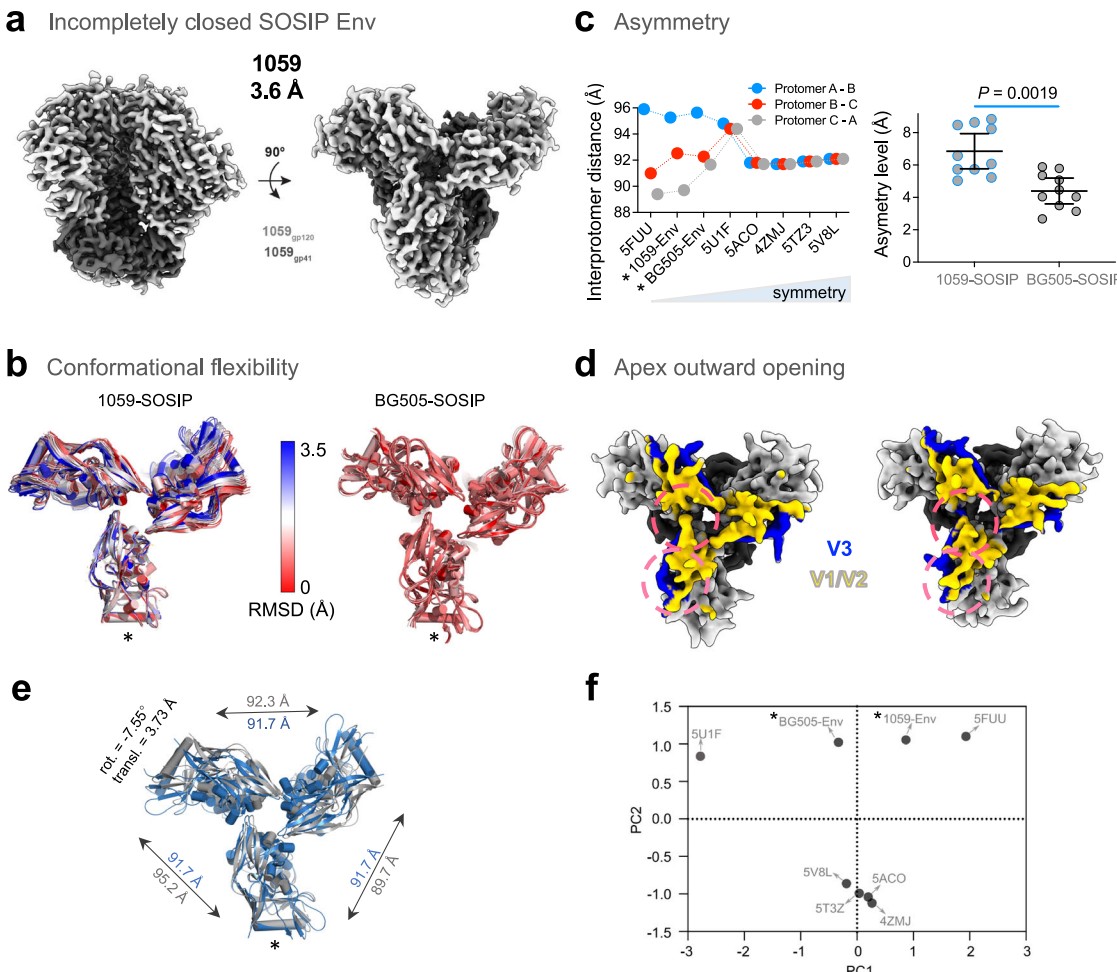

**Fig. 2 | Cryo-EM structure and conformation of incompletely closed SOSIP Envs of a transmitted/founder HIV-1 strain (1059). a** Side and top views of unliganded incompletely closed 1059-SOSIP structure at 3.6 Å. Structural determinants of incompletely closed SOSIP: **b** Conformational flexibility. Overlay of 10 rigid fitted sub-class models of 1059 or BG505 SOSIPs purified and analyzed under similar conditions. Color bar indicates the C-alpha root mean square deviation between those structures. We used one protomer (noted with an asterisk) for superposition and calculated the C-alpha root the mean deviation between the structures. Number of particles used, and resolution of each sub-class model are provided in Supplementary Table 5. **c** Asymmetry. We measured the inter-protomer distances between residues 343 of gp120 α2 helix of different protomers in each structure to assess asymmetry. Left, interprotomer distances of different soluble Envs, including two asymmetric (PDB: 5FUU and 5U1F; refined with C1 symmetry) and four symmetric structures (PDB: 4ZMJ, 5ACO, 5TZ3 and 5V8L; refined with C3 symmetry except the 5V8L). Right, statistical analysis of difference in asymmetry levels between 1059 and BG505 SOSIPs calculated from 10 respective subclasses for each SOSIP. Asymmetry was calculated based on deviation from inter-protomer distances (see Methods). Data are distribution of distances, and mean ± 95% confidence interval. *P*, two-tailed *P* value of Mann–Whitney *U* test. **d** Mobility. CryoSPARC 3D variability analysis revealed disruption of the trimer apex and outward movement of the V1/V2 loop (see Supplementary Movies, principal component 3 motion). Two snapshots are shown where the V1/V2 loop (yellow) is shifting outward from the trimer (pink circles). **e** Unliganded, incompletely closed 1059-SOSIP structure (colored gray) overlaid on the structure of a perfectly symmetric unliganded HIV-1 Env SOSIP crystal structure (PDB: 4ZMJ; colored blue). We evaluated 1059-SOSIP asymmetry from misalignment of the other two protomers of the 1059-SOSIP (gray) with the corresponding protomers of the symmetric reference structure (blue). **f** Principal component analysis of interprotomer distances in different soluble Envs. *, protomer used for superposition (**b**, **e**) and SOSIP structures solved in this study by cryo-EM and refined with C1 symmetry (**c**, **f**).

To study how N6 recognizes more open Env conformations, we measured N6 binding to 1059-SOSIP that was immobilized by 17b, which stabilizes open Env conformation. N6 binding to 17b-immobilized 1059-SOSIP was comparable to N6 binding to 1059-SOSIP that was immobilized by either the control 2G12 or 10-1074 antibodies, which exhibits no conformational preference. N6 also bound glutaraldehyde-crosslinked 1059-SOSIP as efficiently as non-crosslinked 1059-SOSIP (Supplementary Fig. 11). Notably, N6 recognized 17b-bound 1059-SOSIP Envs and HIV-1$_{JRFL}$ intermediate Envs expressed on the cell surface more efficiently than VRC01, 3BNC117, and VRC03 (Fig. 4g and Supplementary Fig. 12c). Efficient recognition of different Env conformations by N6 was consistent with tighter binding to 1059 gp120 (which is completely open/accessible) compared to the binding of other CD4bs bnAbs to 1059 gp120. Moreover,

with strong binding of different Env conformations and in contrast to VRC01, 3BNC117 and VRC03 bnAbs, N6 efficiently neutralized HIV-1 with tightly closed (BG505) and incompletely closed (1059) Envs (Fig. 4h). To better understand how N6 recognizes different Env conformations, we separately adapted in vitro two HIV-1 strains to the presence of N6 (Fig. 4i, j). Each strain carried Envs that prefer to be in a specific conformation: HIV-1$_{NL4-3}$ (CH040) uses the incompletely closed CH040 Envs to enter target cells whereas HIV-1$_{BaL}$ is a lab-adapted virus and uses more open Envs to mediate HIV-1 entry into cells. The two viral strains developed resistance to N6 that differed in kinetics and pattern (Fig. 4j and Supplementary Fig. 13). Development of resistance to HIV-1$_{NL4-3}$ (CH040) was delayed compared to HIV-1$_{BaL}$ and isolated, single resistant clones showed only low or no resistance to N6 in a single-round infection assay. In contrast, N6 resistance of

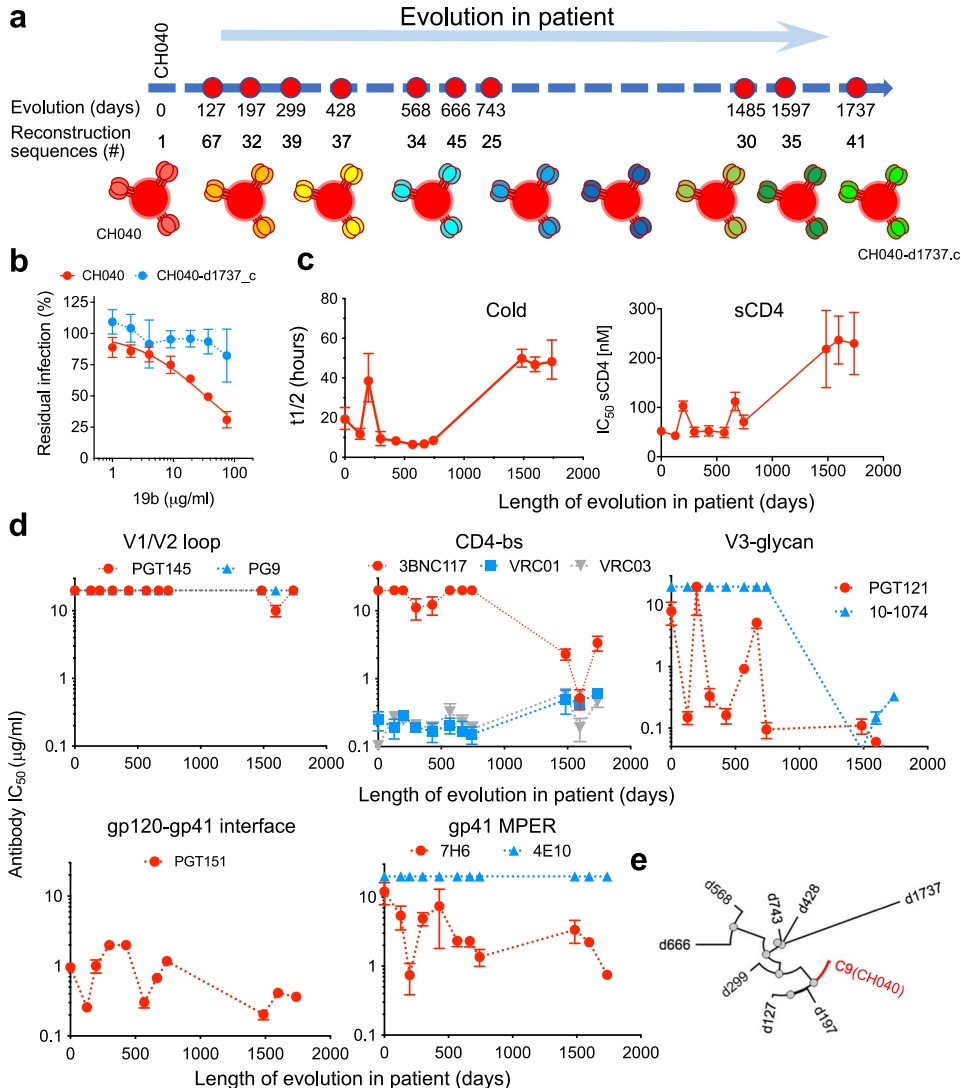

**Fig. 3 | Reconstructing the in vivo evolutionary pathway of a multi-bnAb resistant T/F Envs (CH040) that partially expose the gp120 V3 loop. a** A scheme of 10 time points during evolution of CH040 Envs in infected individual in which available sequences were used to determine consensus *env* sequences and to reconstruct consensus-based pseudoviruses. **b** Sensitivity of HIV-1 pseudotyped with T/F CH040 and CH040-d1737_c, which represents consensus of viral population that evolved by day 1737, to 19b antibody that recognizes exposed V3-loop in open Env conformation. Data are mean values ± s.d.; *n* = 4 technically independent experiments (two separate experiments each with two replicates; data of CH040 is identical to data shown in Fig. 1b and experiment was independently repeated 3 times). **c** Sensitivity of reconstructed consensus pseudoviruses to cold and sCD4.

Data are fitted $t_{1/2}$ (cold) values ± 95% confidence intervals; *n* = 12 technically independent experiments (three separate experiments each with four replicates); and fitted IC$_{50}$ values ± s.e. (sCD4); *n* = 2 biologically independent experiments, each with 4–6 replicates. All curves used for the analysis are shown in Supplementary Fig. 9. **d** Sensitivity of reconstructed consensus pseudoviruses to bnAbs targeting different sites of Env vulnerability. Data are fitted IC$_{50}$ values ± s.e.; *n* = 4 or 8 technically independent experiments (two separate experiments each with either two or four replicates). **e** Phylogenetic tree of CH040 Env evolution in the infected individual based on consensus *env* sequences (generated by NGPhylogeny.fr; Nucleic Acids Research, 47, W260–W265, 2019). Source data are provided as a Source Data file.

HIV-1$_{BaL}$ was robust and resistant clones exhibited different degrees of resistance to N6 in a single-round assay. Several HIV-1$_{BaL}$ clones exhibited complete resistance to N6 up to concentration of 10 µg/ml (Fig. 4j). We identified higher number of mutations in HIV-1$_{BaL}$ Env clones compared to HIV-1$_{NL4-3}$ (CH040) and, notably, clones from both strains contained one or more changes in gp41, which are substantially distant from N6 binding site. Moreover, most adapted HIV-1$_{BaL}$ clones showed altered sensitivity to cold and to Env ligands that recognize internal epitopes, suggesting involvement of conformational changes in the resistance phenotype (Supplementary Fig. 13). Identified changes were also associated with the development of global resistance to all CD4bs bnAb tested (Fig. 4j).

Resistance of 1059 Envs to most CD4bs bnAbs highlights one potential mechanism by which incompletely closed Env conformation can escape bnAb neutralization. In this context, we analyzed the HIV-1 Envs from the recent first-in-humans antibody-mediated prevention (AMP; HVTN 704) trial, in which the CD4bs bnAb VRC01 was administered to healthy individuals at risk as a potential prevention modality[42]. VRC01-resistant Envs were isolated from VRC01-treated individuals who acquired HIV-1 despite the presence of VRC01, and also from individuals in the placebo arm. We tested the sensitivity of Envs isolated in this trial to ligands and cold and detected a statistically significant correlation between resistance to VRC01 and hypersensitivity to cold ($P = 0.001$), which is associated with more open Env conformations (Fig. 5a, b and Supplementary Table 8). Moreover, some VRC01-resistant Envs were susceptible to specific internal-epitope antibodies such as 39 F (binds gp120 V3), E51 (binds gp120 CCR5-bs), and 246-D (binds gp41) in comparison with VRC01-sensitive Envs although, as expected, not every

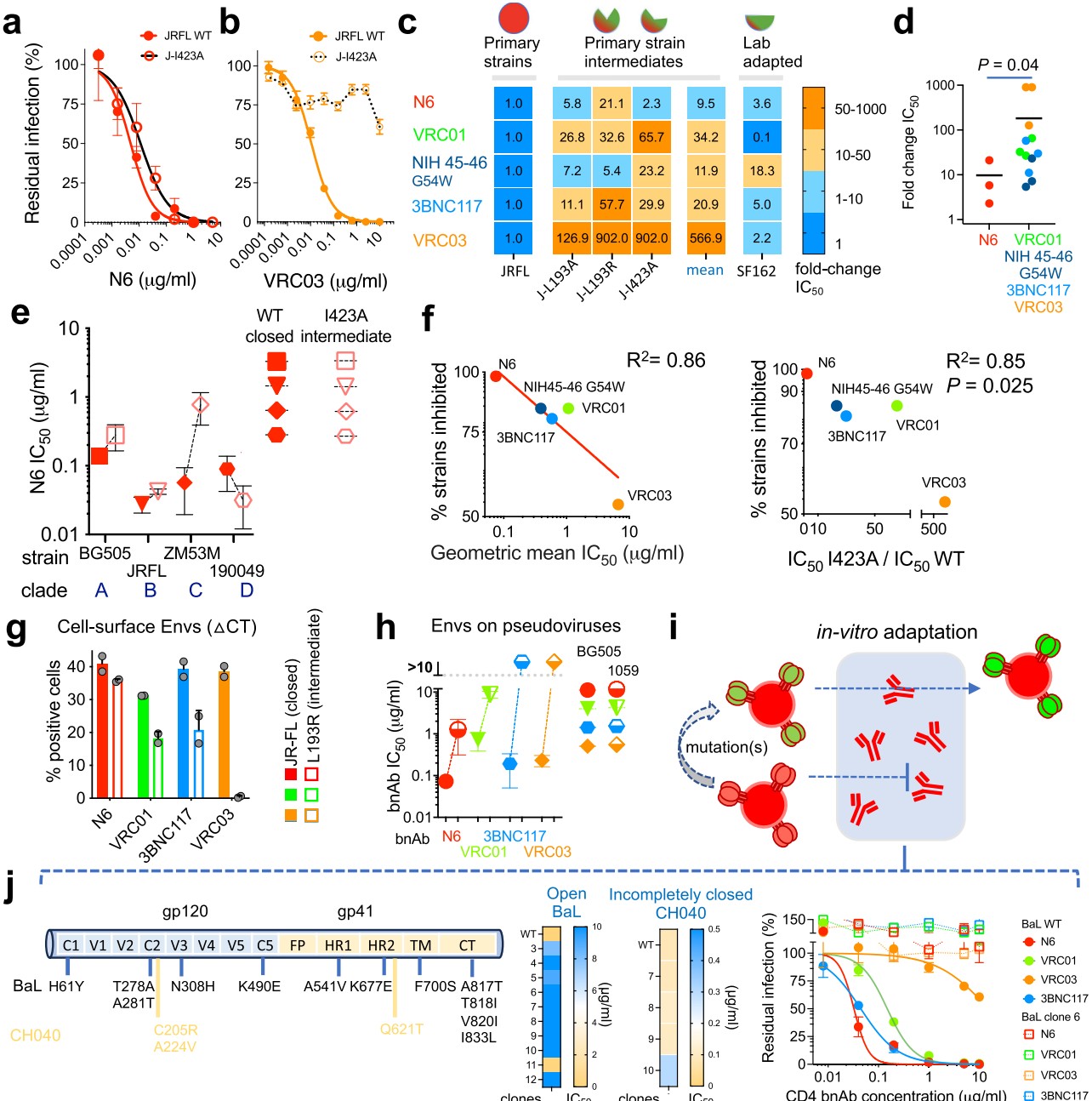

**Fig. 4 | Relationship between neutralization of multiple Env conformations, potency, and breadth of CD4bs bnAbs.** Sensitivity of HIV-1 pseudotyped with WT JRFL (closed), and JRFL I423A (open intermediate) Envs to N6 (**a**) or VRC03 ((**b**); data were adapted from Herschhorn et al., Nat Commun 2017) bnAbs. **c** Fold-change of neutralization sensitivity of pseudoviruses with closed (JRFL WT), intermediate, and lab-adapted (SF162) Env conformations to different CD4bs bnAbs. JRFL Env variants were selected based on Env sensitivity to ligands recognizing open conformation (refs. 7,25). **d** Statistical analysis of HIV-1$_{JRFL}$ Env intermediate recognition by different CD4bs bnAbs. Values are fold change of fitted IC$_{50}$ values (from **c**; $n = 2–4$ independent experiments for HIV-1$_{JRFL}$ Env intermediates; and 2 technical replicates for SF162), line is the mean IC$_{50}$ fold change; *P*, two-tailed *P* value calculated by Mann–Whitney U test. **e** Sensitivity of HIV-1 pseudotyped with the closed (WT) and intermediate (I423A) Env conformations of strains from diverse clades to N6. **f** Relationship between the exposure of internal epitopes resulted in VRC01 resistance (Fig. 5c, d, and Supplementary Fig. 14). Principal component analysis of Env sensitivity clustered VRC01 sensitivity opposite to E51, 17b, 246-D and cold sensitivities (Fig. 5d). Of note, 1059 is resistant to VRC01 and exhibited

breadth of CD4bs bnAbs and their potency (left), and bnAb breadth and efficiency to neutralize different HIV-1$_{JRFL}$ Env conformations (right). *P*, two-tailed *P* value of Pearson correlation. **g** Flow cytometric analysis of bnAb binding to different conformations of HIV-1$_{JRFL}$ Env expressed on cell surface. ΔCT, cytoplasmic tail deleted. **h** Sensitivity of BG505 and 1059 pseudoviruses to CD4bs bnAbs. **i** Scheme of the experiment to adapt HIV-1 in vitro to N6. **j** Analysis of adapted clones of HIV-1$_{NL4-3}$ (CH040), and the lab-adapted HIV-1$_{BaL}$ to N6. Left, major changes in adapted stains compared to wild type; Middle, heatmap of IC$_{50}$ of adapted clones; Right, sensitivity of adapted HIV-1$_{BaL}$ clone 6 to CD4bs bnAbs in single-round infection assay. Data in (**a**, **b**, **e**, **g**, **h**, **j**) are mean values ± s.e.m (**a**, **b**, **j**), fitted IC$_{50}$ values ± s.e. (**e**, **h**) and mean values ± s.d. (**g**) $n = 2–3$ independent experiments, each performed in duplicate (**a**, **b**, **e**, **j**) and data are representative of $n = 2$ independent experiments, each performed in duplicate (**g**). Source data are provided as a Source Data file.

similar hypersensitivity to cold, and to E51 antibody (Supplementary Figs. 2, 3). With one exception (of a single VRC03-sensitive strain), all VRC01-resistant Envs were also resistant to VRC03 and 3BNC117 (IC$_{50}$ > 5 µg/ml) that, similarly to VRC01, exhibited reduced

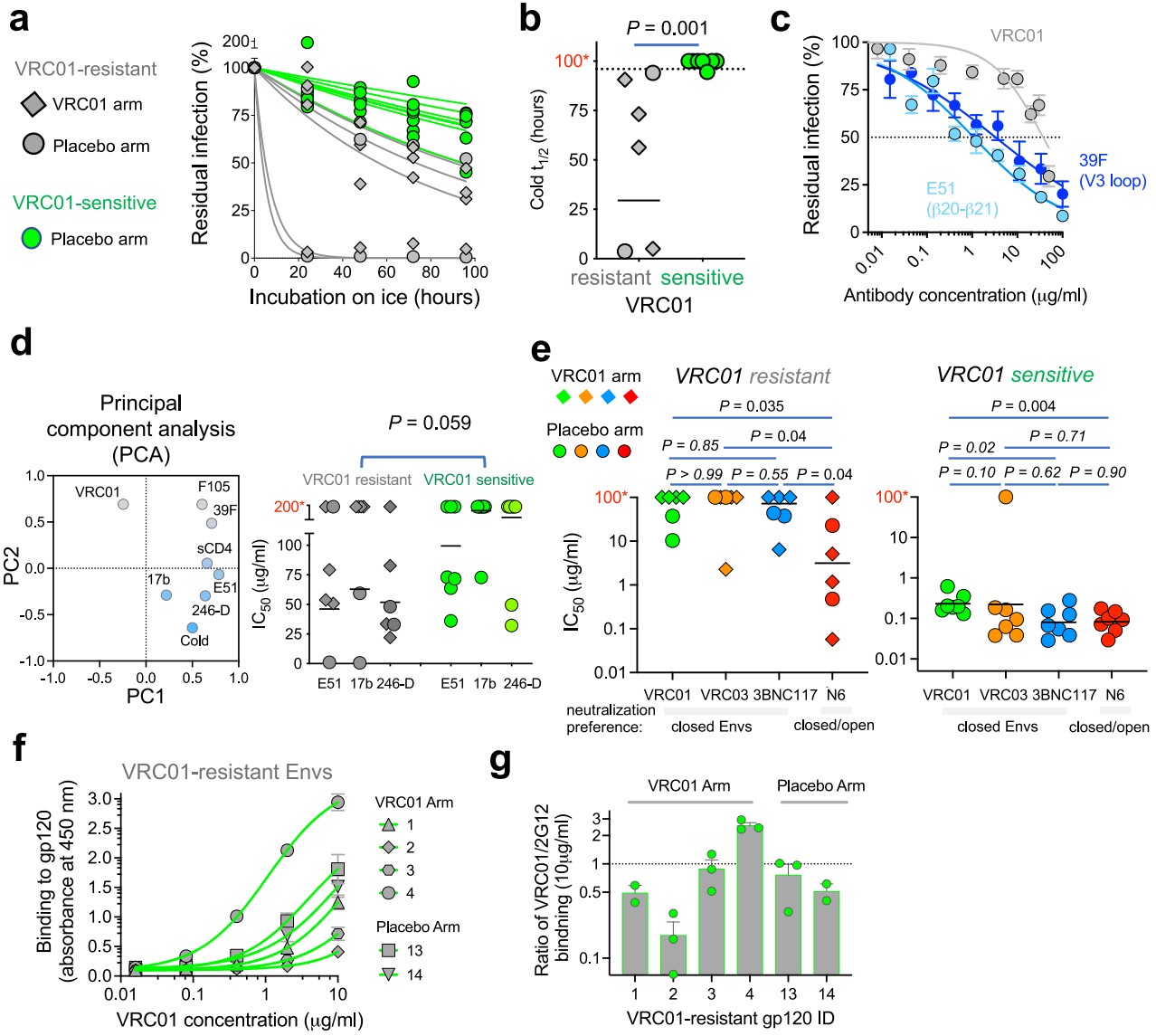

**Fig. 5 | Analysis of HIV-1 Envs isolated from the antibody mediated prevention (AMP) trial (HVTN 704). a** Sensitivity to cold exposure of VRC01-resistant Envs, isolated from the VRC01 or placebo arms, and of VRC01-sensitive Envs, isolated from the placebo arm. **b** Statistical analysis of difference between cold sensitivity of VRC01-resistant and VRC01-sensitive Envs. We compared half-life on ice after fitting zero-order decay curves to the residual infection data (from **a**). Dotted line, maximal experimental time (96 h) tested. **c** Sensitivity of single VRC01-resistant Envs (H704_1835_150_RE_p002s_2484A) to VRC01 and internal-epitope antibodies. **d** Left - PCA of 6 VRC01-resistant and 7 VRC01-sensitive Envs clustered VRC01 sensitivity opposite to E51, 17b, 246-D and cold sensitivities. All Envs were resistant to 19b and 697-30D and thus these antibodies were not included. Right - statistical analysis of the difference between the sensitivity of VRC01-resistant and VRC01-sensitive Envs to 3 Abs (E51, 17b, and 246-D) identified by PCA. IC$_{50}$s were calculated from dose-response curves and no differences were identified for all other Env ligands or for comparison of VRC01-resistant and VRC01-sensitive for each antibody separately (Supplementary Fig. 14). **e** Sensitivity of VRC01-resistant (left) and VRC01-sensitive (right) Envs to CD4bs bnAbs. **f** Binding of VRC01 to soluble AMP gp120s measured by ELISA. **g** 2G12-normalized VRC01 binding to AMP gp120s. $t_{1/2}$ or IC$_{50}$ values higher than the maximal time/concentration tested (96 h (**b**), 100 μg/ml (**d**) and 50 μg/ml (**e**)) were set to an arbitrary 100 h (**b**), 200 μg/ml (**d**) and 100 μg/ml concentrations (**e**), which are labeled (red letters and asterisks) on the Y-axis. In (**b**, **d**, **e**), *P* is two-tailed *P* value of Mann–Whitney U test. Data are mean values (**a**, **d**); mean ± s.e.m. (**c**, **f**, **g**); calculated half-life on ice (**b**); or fitted IC$_{50}$ values (**e**); *n* = 2–4 (**a**, **b**), 3 (**c**), or 2–5 (**d**, **e**, **f**, **g**) independent experiments, each performed with 2–4 replicates. Source data are provided as a Source Data file.

neutralization activity against Env intermediates (Fig. 5e)[7,25]. Despite resistance to VRC01, gp120 of most resistant AMP Envs bound VRC01, based on enzyme-linked immunosorbent assay, at concentrations that exhibit no significant effect on viral entry of these resistant strains (Fig. 5e–g), suggesting interference with VRC01 access to its epitope on the native Envs on HIV-1 virions. In contrast to VRC01, VRC03 and 3BNC117, the ultra-broad N6 bnAb, which efficiently neutralizes different Env conformations, blocked the entry of 50% of the VRC01-resistant strains at IC$_{50}$ < 1.2 μg/ml (Fig. 5e). With one exception, all VRC01-sensitive Envs were sensitive to all other CD4bs bnAbs (Fig. 5e). Overall,

these observations suggest that N6 directly binds open, intermediate, incompletely closed, and closed Env conformations, and this ability allows N6 to potently neutralize diverse Env conformations that exist in different primary HIV-1 strains. Env conformational flexibility may allow some HIV-1 strains to escape CD4bs bnAbs that exhibit reduced activity against Env intermediates (e.g., VRC01).

## Discussion

Here we present evidence that the functional Envs of some T/F HIV-1 strains sample incompletely closed conformations. Six out of thirteen

T/F strains exhibited partial exposure of internal Env epitopes that was apparently compatible with HIV-1 transmission. We selected highly infectious T/F Envs to ensure robust and reproducible results; the levels of viral infectivity were unrelated to internal-epitope exposure, sCD4 or cold sensitivity (Supplementary Fig. 3e), and thus our approach likely recapitulates a general phenotype of Env function. Our results (6/13; 46%) may underestimate the true number of incompletely closed Envs as our ability to identify this phenotype is limited by available antibodies and the presence of their epitopes in all strains. Incompletely closed T/F Envs were hypersensitive to cold, which is an antibody-independent method associated with Env opening[7,18,32]. The 17b and E51 antibodies bind to epitopes that overlap with the coreceptor-bs and contact the β20-β21 element of gp120, which regulates Env transitions between different conformations[25,43]. Changes in two Env resides, K421 and Q422, main 17b and E51 contacts typically not exposed in primary isolates, lead to Env opening and hypersensitivity to ligands that recognize internal epitopes[25]. Thus, weak-moderate recognition of 1059, SC45 and WEAU Env trimers expressed on the surface of HIV-1 by 17b and E51 (Supplementary Fig. 2) suggest that the β20-β21 regulatory switch in some incompletely closed Envs is oriented in favorable downstream configuration. Cryo-EM structure of the unliganded 1059-SOSIP provides insights into an incompletely closed Envs. Trimer asymmetry, increased conformational flexibility, and scissoring motions provide the long-sought structural evidence for putative "breathing" ability of HIV-1 Envs. Our experimental finding is consistent with previous molecular dynamics simulations of a fully glycosylated unliganded HIV-1 BG505 SOSIP Env trimer, in which protomer flexing about the trimer axis (i.e., scissoring motion) coupled with steric occlusion by glycans was found to restrict access to the CD4bs[44] and with studies that associated trimer asymmetry with initial response to CD4[45,46]. V1/V2 movement such as the one induced by amino acid changes in V1/V2 (e.g., L193A) can lead to HIV-1 resistance to VRC01, VRC03, and 3BNC117 bnAbs[7,18,25], similar to the resistance profile that we measured for 1059 (Supplementary Figs. 2, 4). Of note, 1059 Envs provide the highest (22.6%) T cell epitope coverage among T/F Envs and has been tested as a preferred Env immunogen in non-human primates[47] and *humans* (HVTN 106; NCT02296541).

Post transmission evolution of the incompletely closed CH040 T/F Envs towards more closed conformation is consistent with the known pressure of strain-specific neutralizing antibodies that are developed in individuals infected with HIV-1. We hypothesize that generation of incompletely closed Envs potentially increases the number of alternative pathways to adapt to a closed Env conformation in comparison with maintaining the closed Env conformation at all times, which is expected to limit the allowed changes and possible pathways to escape neutralizing antibodies. This hypothesis is consistent with the robust, in vitro evolution of the lab adapted HIV-1$_{BaL}$ with more open Envs to develop resistance to N6. Additionally, temporal existence of more open Env conformations may facilitate the infection of cells that express low levels of CD4[7,25] and Env interactions with CD4 receptors of additional species[48].

Our results suggest that incompletely closed Envs are a heterogeneous collection of similar but not identical Env conformations. The gp120 V3 region is the first and most common element to be exposed on the Envs of T/F isolates and of intermediate variants generated in vitro, consistent with previous reports of V3 metastability[20,49]. We found that alternative and multiple routes for Envs to exist in incompletely closed conformations can potentially affect bnAb sensitivity in different ways. For example, CH040 Envs expose the V3 loop according to 19b sensitivity and are resistant to V3-glycan PGT121 and 10-1074 bnAbs (Fig. 3). In contrast, WEAU Envs are sensitive to different antibodies that target internal epitopes as well as to most bnAbs tested (Fig. 1 and Supplementary Fig. 2). Thus, strain-specific heterogeneity of the incompletely close Env conformations may affect bnAb sensitivity.

HIV-1 Env resistance to multiple CD4bs bnAbs among VRC01-resistant strains from the AMP trial highlights a potential mechanism by which incompletely closed Env conformation can escape neutralization by several bnAbs. These results underlie potential pathways of global HIV-1 resistance to VRC01-like bnAbs and superiority of CD4bs bnAbs that neutralize multiple Env conformations. Thus, bnAbs recognizing diverse Env conformations may have superior breadth and some V3-glycan bnAbs neutralize different Env conformations because they recognize conserved glycans that are unaffected by Env transitions. Similarly, the gp120-gp41 interface bnAb 8ANC195 recognizes partially open and closed conformations and prevents full opening of Env trimer by binding to protein residues and N-linked glycans (ref. 50). However, the breadth of these bnAbs against diverse HIV-1 strains is limited by heterogeneous glycosylation patterns in cells. In contrast, recognition of highly conserved amino acids and open Env conformations by MPER and N6 bnAbs are associated with ultra-broad neutralization. Thus, affinity maturation of N6 optimized Env binding to a maximum number of highly conserved amino acid residues, all or most of which are accessible on different Env conformations; this strategy allows N6 to neutralize 98% of HIV-1 isolates[29]. Our data suggest that there are many alternative pathways by which HIV-1 Envs can become incompletely closed or even partially open and only some of these alternatives will lead to bnAb resistance. Importantly, while VRC01, VRC03 and 3BNC117 bnAbs are still mostly effective against Envs of lab-adapted HIV-1 strains that evolved in vitro to become fully open, N6 is effective against these strains and, in addition, primary HIV-1 Envs that are stabilized in intermediate states (e.g., JRFL I423A). Thus, additional constraints are imposed to restrict opening of Envs of primary strains compared with opening of lab-adapted Envs, which may be fully accessible to antibodies. N6 will likely be more effective than VRC01, VRC03, and 3BNC117 against primary Envs as they evolve to incompletely closed conformations (e.g., 1059). Two additional broad and potent CD4bs bnAbs, Ab1-18 and N49P7, have been recently identified[51,52] and may have a similar mode of action.

Our study has several limitations. We tested a relatively small number of Envs from two separate and independent panels (T/F and AMP Envs) and, thus, our observations may not accurately represent the proportion of incompletely closed Envs in the viral population. Our mechanistic insights are partially based on structure of SOSIP Env. Although recent studies of SOSIP conformational changes were consistent with data driven form Envs on virions[46,53], the use of SOSIP may have underestimated Env opening as this soluble format was originally engineered to capture closed, prefusion Env conformation. We analyzed symmetry of only two SOSIPs purified via GNL under identical conditions; we expect that analysis of SOSIP Envs of additional HIV-1 strains, which are planned for future studies, will reveal a range of asymmetries that may relate to the ability of different Envs to "breathe". Despite these limitations, our results clearly refine the current knowledge of HIV-1 conformation heterogeneity and provide evidence that not every HIV-1 Env from ~37 million different individuals infected with HIV-1 adopts a single identical closed Env conformation. Maintenance of a closed Env conformation is balanced by Env conformational flexibility within the circulating HIV-1 population and during Env synthesis and evolution in patients. The existence of incompletely closed Env conformation impacts the development of next generation bnAbs for immunotherapy, the design of Env immunogens and the understanding of HIV-1 Env function and inhibition. Our results, together with the outcome of several clinical trials[42,54,55], suggest prioritizing N6-like bnAbs as preferred candidates for future trials using CD4bs bnAbs. In parallel, an alternative vaccine approach in which a combination of tightly- and incompletely-closed Env immunogens are presented to the immune system may be an efficient way to mount a broad antibody response against diverse HIV-1 strains, some of which display incompletely closed Envs[56]. In this context, the HVTN 106 trial (NCT02296541) has tested the immune response of

DNA-based 1059 Env immunogen in *humans* with the aim of induction both efficient B and T cell Env responses and upcoming results will further inform the development of new approaches using incompletely-closed Env immunogens.

# Methods

## Cell lines

293 T cells were purchased from the American Type Culture Collection (ATCC) and the TZM-bl cells were obtained from the NIH AIDS Reagent Program. Expi293F/293F cells for protein expression were purchased from Gibco (ThermoFisher Scientific). Cf2Th-CD4/CCR5, Cf2Th-CCR5, and Cf2Th-CD4/CXCR4 were generated in the laboratory of Joseph Sodroski. Cell lines were not authenticated and were tested negative for contamination with mycoplasma.

Expi293F/293F cells were maintained at 37 °C and 8% $CO_2$ in Expi293/293F freestyle Expression medium (Gibco; ThermoFisher Scientific) with continuous shaking at 110–130 rpm. 293 T and TZM-bl cells were grown in Dulbecco's Modified Eagle Medium (DMEM) containing 10% Fetal Bovine Serum (FBS), 100 µg/ml streptomycin and 100 units/ml penicillin. Cf2Th-CD4/CCR5 and Cf2Th-CD4/CXCR4 cells were grown in the same medium supplemented with 400 µg/ml G418 and 200 µg/ml hygromycin B (both from Invitrogen, ThermoFisher Scientific). Cf2Th-CCR5 cells were grown in the same medium supplemented with 400 µg/ml G418.

## Plasmid construction

Expression plasmids for T/F Envs were obtained from the NIH AIDS Reagent Program. HIV-1$_{JRFL}$ mutants were previously generated[7,25]; HIV-1$_{CH040}$ K160N and HIV-1$_{CH040}$ E332N *env* mutants were generated by site directed mutagenesis using the following primers:

CH040 E332N-F: 5′-ggagacataagaaaagcatattgtaatattaatggaacagaat ggcatagc-3′

CH040 E332N-R: 5′-gctatgccattctgttccattaatattacaatatgctttctttatg tctcc-3′

CH040 K160N-F: 5′-gaagggagaagtaaaaaactgttctttcaatatcaccacaga cataa-3′

CH040 K160N-R: 5′-ttatgtctgtggtgatattgaaagaacagtttttttacttctccct tc-3′

Correct DNA sequences were verified by Sanger sequencing. CH040 *env* variants based on consensuses sequences of viruses that evolved over time in patient were generated by gene synthesis (Gene Universal, Newark, DE). Codon-optimized 1059-SOSIP was synthesized and cloned into pTwist-CMV_Betaglobin_WPRE_Neo vector (by Twist Biosciences, San Francisco, CA); the vector was further modified to replace the natural signal peptide with the tissue plasminogen activator (TPA) signal peptide and to introduce stabilizing mutations (by Gene Universal, Newark, DE). Genes for expression of monomeric gp120s were codon-optimized and cloned into a pTwist-CMV_Betaglobin_WPRE_Neo vector (by Gene Universal, Newark, DE). The vector contained an 8-Histide tag at the C-terminus to facilitate purification by Ni-NTA chromatography.

## Production of recombinant HIV-1 expressing luciferase

We produced viruses by cotransfecting 293 T cells with three plasmids: an envelope-expression plasmid, pHIVec2.luc reporter plasmid and psPAX2 packaging plasmid (catalog number 11348, NIH AIDS Reagent Program) in a ratio of 1:6:3 using either Effectene (Qiagen) or calcium phosphate, as previously described[57,58]. After a 48 h incubation, the cell supernatant was collected and centrifuged for 5 min at 600–900 × g at 4 °C. The amount of p24 in the supernatant was measured using the HIV-1 p24 antigen capture assay (catalog number 5421, Advanced BioScience Laboratories or in-house p24 assay) and the virus-containing supernatant was frozen in single-use aliquots at −80 °C.

## Viral infection assay

A single-round infection assay was performed as previously described[7,57,59,60]. Briefly, HIV-1 Env ligands (antibodies and sCD4) were diluted in DMEM and thirty microliters of each tested concentration were manually dispensed into a well (with 2–3 replicates for each concentration) of a 96-well white plate (Greiner Bio-One, NC). Thirty microliters of viruses pseudotyped with specific Envs were then added, and after a brief incubation at room temperature, thirty microliters of $1.7 \times 10^5$ Cf2Th-CD4/CCR5 target cells/ml in DMEM were added. We used either 2 ng of p24 of virus preparations or tittered the viruses on Cf2Th-CD4/CCR5 cells and used 1–3 million relative light units (2 s integration using Centro LB 960 or Centro XS$^3$ LB 960 luminometer) on all experiments. After 48–72 h incubation, the medium was aspirated and cells were lysed with 30 µl of Lysis Buffer. The activity of the firefly luciferase, which was used as a reporter protein in the viral assay, was measured with a Centro LB 960 (or Centro XS$^3$ LB 960) luminometer (Berthold Technologies, TN, USA).

To assess the effect of exposure to cold on virus infectivity, single-use aliquots of recombinant virus preparations were thawed at 37 °C for 1.5 min at indicated intervals and then incubated on ice for different time periods. At the end of the incubation, equal amounts of recombinant viruses were added to Cf2Th-CD4/CCR5 target cells to assess infectivity.

## Calculation of half maximal inhibitory concentration (IC$_{50}$) and infectivity decay on ice

Dose response curves of viral infection assay were fitted to the four-parameter logistic equation using Prism 9 program (GraphPad, San Diego, CA) after adding the equation to the program; IC$_{50}$ values and the associated s.e. are reported[61–63]. Dose response curves of virus infectivity decay on ice were fitted to one phase exponential decay using the Prism 7 program. Number of experiment repeats and replicates are provided in Figure Legends.

## in vitro adaptation of HIV-1 to N6

HIV-1$_{BaL}$ stocks were provided by the NIH HIV Reagent Program (catalog number ARP-510) and HIV-1$_{NL4-3}$ (CH040) stocks were prepared by transfection of 293 T cells with a molecular clone and collecting the virus-containing supernatant after 48 h. p24 concentration was measured by p24 ELISA kit (catalog number XB-1000, Xpress Bio). HIV-1 preparation containing 30 ng (HIV-1$_{BaL}$) or 50 ng (HIV-1$_{NL4-3}$ (CH040)) p24 were used to infect SupT1-CCR5 cells in 12-well or 6-well plates by spinoculation (1200 × g for 2 h at 25 °C)[64]. HIV-1 replication was monitored by measuring p24 concentration in the culture supernatant[65] and the virus was passaged to fresh SupT1-CCR5 cells at peak replication (typically up to 28 days). During passage 1, HIV-1 strains were allowed to replicate without N6 for 10–18 days to increase the viral population diversity. In passage 2, HIV-1 was incubated with 0.5 µg/ml of the CD4bs bnAb N6 and the concentration was increased to 3 µg/ml during passage 3. At the end of each passage, the genomic DNA was isolated and the strain-specific HIV-1 *env* gene within the provirus was amplified by PCR using the previously published Envout primers[27]:

Env5out 5′-TAGAGCCCTGGAAGCATCCAGGAAG-3′

Env3out 5′-TTGCTACTTGTGATTGCTCCATGT-3′

For a second round PCR, we extended published Envin primers and added sequences that overlap with a pcDNA-based Env expression plasmid (p1059_09.A4.1460; NIH HIV Reagent Program) to allow Gibson Assembly of the *env* gene into the pcDNA-based expression vector. The DNA sequence of complete primers used are (sequence added to allow Gibson assembly are shown in lower case and italic):

Env5inG  5′-*gttaagcttggtaccgagctcggatcc*TTAGGCATCTCCTATGG-CAGGAAGAAG-3′

Env3inG 5′-*taccttcgaaccgcgggccctctaga*GTCTCGAGATACTGCTCC-CACCC-3′

*env* gene of adapted strains was cloned into the p1059_09.A4.1460 Env-expressing plasmid (replacing the 1059 env gene), analyzed by Sanger sequencing, and the plasmid was used to generate single-round pseudovirus clones of adapted HIV-1 strains. For each passage, the DNA sequence of at least 10 independent clones was analyzed by Sanger sequencing and compared to WT *env* sequences.

### Cell-cell fusion assay

Cell-cell fusion was monitored as previously described[66] but using HIV-1 tat as a transactivator and TZM-bl cells as reporter (target) cells. Briefly, 500,000 293 T cells in each well of a six-well plate were co-transfected with HIV Envs and HIV-1 tat expression plasmids at a ratio of 6:1 (total 2 μg). After 48 h, the cells were detached with 5 mM EDTA/PBS, and 10,000 cells were added to each well of TZM-bl reporter target cells that were pre-seeded (10,000/well) the day before. Cells were incubated for the specified time and then lysed; luciferase activity was measured and used to evaluate the extent of fusion.

### Flow cytometry

Binding of CD4bs bnAbs to HIV-1$_{JRFL}$ Envs expressed on the surface of 293 T cells was analyzed by flow cytometry as previously described[7]. Briefly, 293 T cells were transfected with an Env-expressing plasmid for expression of either HIV-1$_{JRFL\_WT}$ or HIV-1$_{JRFL\_L193R}$ Envs, which adopts a more open Env conformation[7]. Cytoplasmic tail of both expressed Envs was deleted (ΔCT) to allow a high level of Env expression to reliably measure CD4bs bnAb binding. Forty-eight hours after transfection, cells were detached with 5 mM EDTA/PBS, washed, and 300,000 cells in 100 μl 5% FBS/PBS were incubated with 0.01 μg/ml of indicated bnAbs for 30 min. Cells were washed twice and incubated with Allophycocyanin (APC)-conjugated F(ab')2 fragment donkey anti-human IgG antibody (1:100 dilution; catalog number 709-136-149; Jackson ImmunoResearch Laboratories) for 30 min. Cells were washed twice and analyzed by CytoFLEX flow cytometer (Beckman Coulter) using CytExpert software. All procedures were performed at room temperature.

### Protein expression and purification

**sCD4**. Plasmid containing a codon-optimized sCD4 gene was transfected into 293F cells using 293fectin transfection reagent (Invitrogen, Thermo Fisher Scientific Inc.). Transfected cells were grown for 3–6 days at 37 °C and 8% $CO_2$ with continuous shaking and the secreted sCD4 containing a 6-histidine tag at its C-terminus, was purified from the culture supernatant using Ni-NTA chromatography[67]. Binding of sCD4 to the Ni-NTA column was weak and therefore to allow efficient binding we did not add a low concentration of imidazole to the supernatant prior to sCD4 purification.

**Monomeric gp120**. 293F cells were transfected with a gp120-expressing plasmid using Turbo293 transfection reagent (Speed Biosystems; Gaithersburg, MD) and grown for 3–5 days in a tissue culture incubator at 37 °C, 8% $CO_2$ with continuous shaking. Culture supernatant containing soluble gp120 glycoprotein was then clarified by centrifugation at $7000 \times g$ for 1 h and filtered using a 0.45 μm vacuum filtering system (VWR). Supernatant was dialyzed against Ni-NTA buffer (50 mM NaH$_2$PO$_4$, 300 mM NaCl; pH 8.0) overnight using a 13,000 MWCO dialysis tube (VWR) and the dialyzed supernatant was loaded on a column of Ni-NTA agarose beads (Qiagen) at 4–8 °C. The column was washed with 500 mM NaCl in phosphate buffered saline pH 8.0 and the gp120 glycoprotein was stepwise eluted with Ni-NTA buffer containing 50–250 mM imidazole (MilliporeSigma) solutions. Elution fractions were analyzed on 8–16% SDS-PAGE (mini-PROTEAN TGX protein gels; Bio-Rad) and fractions containing the gp120 glycoprotein were concentrated followed by buffer exchange to PBS using Vivaspin 6 centrifugal concentrators (30 kDa; Cytiva). Purified gp120 glycoproteins were flash frozen and stored in aliquots at −80 °C.

**SOSIP trimers**. 1059- and BG505-SOSIP were purified by GNL chromatography, which exploits lectin binding to Env surface glycans in a conformation-independent manner, for immunological assays and for determining unliganded SOSIP structures. All preparations were further purified by size-exclusion chromatography.

293 F cells were co-transfected with a SOSIP-expressing plasmid and a human furin-expressing plasmid at a 4:1 ratio and grown for 3–5 days in a tissue culture incubator at 37 °C, 8% $CO_2$ with continuous shaking. Culture supernatant was then clarified by centrifugation and filtered using a 0.2 μm vacuum filtering system (VWR). Filtered culture supernatant was kept on ice, loaded on GNL column (Vector Laboratories) at 4 °C and washed with 2–3 column volumes of 500 mM NaCl in phosphate buffered saline pH 8. SOSIP glycoproteins were eluted with 1 M methyl-α-D-mannopyranoside (from Vector Laboratories or MilliporeSigma), filtered through 0.2 μm filter and concentrated using Vivaspin 6 centrifugal concentrators (30 kDa; Cytiva). Purified SOSIP glycoproteins were then separated on a HiLoad 16/600 Superdex 200 pg size exclusion chromatography column (Cytiva) and fractions corresponding to SOSIP trimers were pooled, concentrated, and stored in aliquots at −80 °C.

### Enzyme-linked immunosorbent assay (ELISA)

We used ELISA to analyze antibody binding to SOSIP trimers as previously described[56]. Briefly, Env-specific capturing antibody JR52 was immobilized in wells of a high binding, flat-bottom 96-well plate (Greiner Bio-One, NC) by adding 0.4 μg of JR52 in 100 μl PBS in each well and incubating the plates overnight at room temperature (RT). Next, the wells were washed 3 times with PBS containing 0.2% Tween-20 (wash solution) and blocked with PBS containing 3% bovine serum albumin (blocking solution) for 2 h at RT. The wells were then washed three times, 0.25–0.5 μg of purified SOSIP trimers in blocking solution were added to test wells and the plate was incubated for 2 h at RT. Wells were washed six times and specified antibodies were added at different concentrations. After 1 h and 30 min incubation, wells were washed six times and 1:5000 dilution of horseradish peroxidase (HRP)-conjugated donkey anti-human IgG (FC specific; Jackson ImmunoResearch Laboratories, West Grove, PA) was added in blocking solution to each well and the plate was incubated for 1 h at RT. Wells were then washed six times and 100 μl of TMB solution (1 ml of 1 mg/ml 3,3,5,5-tetramethylbenzidine (MIlliporeSigma) in DMSO, 9 ml of 0.1 M sodium acetate, pH 5.0, and 2 μl of fresh 30% hydrogen peroxide) were added to each well. After -18-min incubation, the HRP reaction was stopped by adding 50 μl of 0.5 M H$_2$SO$_4$ and optical density at 450 nm was measured using Synergy|H1 microplate reader (BioTek). Binding to 1059-SOSIP open conformation was measured in a similar manner, but we immobilized 17b (or 2G12 or 10-1074 controls) antibody to capture 1059-SOSIP, and used biotinylated N6, which was prepared according to manufacturer's instructions (EZ-Link™ Sulfo-NHS-LC-Biotinylation Kit; ThermoFisher Scientific), followed by streptavidin-HRP to detect N6 bnAb binding. In separate experiments, we used 17b Fab to capture 1059-SOSIP open conformation and detected bnAb binding with HRP-conjugated donkey anti-human IgG as described above. For experiments involving glutaraldehyde cross-linking, wells were incubated with 5 mM glutaraldehyde in DDW for 15 min followed by blocking with 25 mM glycine solution. In some experiments, monomeric gp120s and SOSIPs were immobilized through soluble GNL (Vector Laboratories) and all subsequent steps were performed as described above.

### Western blot

gp120 or SOSIP glycoproteins were separated on 8–16% SDS-PAGE (mini-PROTEAN TGX protein gels; Bio-Rad) and transferred to a 0.45 μm nitrocellulose membrane (catalog number 1620115, Bio-Rad). The membrane was blocked with 5% blotting-grade blocker (catalog number 1706404, Bio-Rad) in PBS (5%MPBS), washed with PBS, and incubated for 1 h on a shaker with serum from a person living with HIV-

1 (de-identified sample; 1:30,000 dilution) and sheep anti-gp120 IgG (1:30,000 dilution; catalog number 288, NIH AIDS reagent program) both diluted in 5%MPBS. After three washes with 0.05% Tween 20 (catalog number 1706531, Bio-Rad) in PBS (TPBS), the membrane was incubated with peroxidase conjugated anti-human IgG (1:10,000 dilution) and anti-sheep IgG (1:10,000 dilution) (Jackson Immunoresearch) in 5%MPBS for 1 h. The membrane was washed three times with TPBS, developed with SuperSignal West Pico PLUS Chemiluminescent Substrate (catalog number 34580, ThermoFisher Scientific), and analyzed using the Odyssey imaging system (LI-COR Biosciences). In some cases, we used serum from a person living with HIV-1 (de-identified sample; 1:10,000 dilution) without sheep anti-gp120 IgG.

### Cryo-EM data collection and processing for unliganded 1059 and BG505 SOSIPs (purified by GNL/SEC)

Sample concentrations used were 2.16 mg/mL for 1059-SOSIP and 3.64 mg/mL for BG505-SOSIP. To prevent interaction of the trimer complexes with the air-water interface during vitrification, the samples were incubated in 0.085 mM n-dodecyl β-D-maltoside (DDM) before vitrification. Samples were applied to plasma-cleaned QUANTIFOIL holey carbon grids (EMS, R1.2/1.3 Cu 300 mesh) followed by a 30 s adsorption period and blotting with filter paper. The grid was then plunge frozen in liquid ethane using an EM GP2 plunge freezer (Leica, 90–95% relative humidity). Cryo-EM data were collected using a 300 kV FEI Titan Krios electron microscope (ThermoFisher Scientific) equipped with a K3 camera (Gatan) and GIF Quantum energy filter (20 eV slit width) operating at 81kx magnification with a pixel size of 1.08 Å. Gatan latitude software was used to collect a total of 13,405 and 14,296 movies for the 1059-SOSIP and the BG505-SOSIP structures, respectively.

The data were processed in cryoSPARCv4.01 (ref. [68]). Movies were aligned using Patch Motion Correction and the non-dose weighted aligned micrographs were used for the CTF correction with PatchCTF Estimation. In the first round of particle selection, particles were picked with a box size of 280 Å using the Blob picker with a circular blob having a maximum diameter of 240 Å and these particles were subjected to multiple rounds of 2D classification. The 2D class-averages that best represent the views of 1059-SOSIP glycoprotein structure were selected as templates for the second round of particle selection using the Template picker. Subsequently, these particles were also subjected to multiple rounds of 2D classification yielding few more hundred particles than the Blob picker. The best 2D class averages were selected and processed by generating multiple classes of ab-initio models and followed by heterogeneous classification. 3D classes corresponding to 1059-SOSIP structure were selected and merged as they represent similar structures with small local changes and performed non-uniform refinement and followed by local refinement with C1 symmetry to yield one final map with global resolution of 3.6 Å (Supplementary Fig. 7) at the gold-standard FSC 0.143 criterion. Remote 3DFSC Processing Server[69] was used to generate 3D FSCs of maps which calculated resolutions using gold-standard FSC 0.143 criterion[69]. Local resolutions of the refined maps were generated using cryoSPARC[68]. Similar procedures were used for the data collection and processing of unliganded BG505-SOSIP. After the data processing, unliganded BG505 SOSIP data set resulted in 975,399 particles refined with C1 symmetry and yielded a final map with global resolution of 3.7 Å (Supplementary Fig. 7) at the gold-standard FSC 0.143 criterion. For conformational flexibility and asymmetry level analysis, both the unliganded 1059 and BG505 SOSIP datasets were subclassified by 3D variability and reconstructed 10 subclasses ranging from resolution of 3.7–4.3 Å (Supplementary Table 5).

### Model fitting

The initial homology model of the 1059-SOSIP Env was constructed using the Modeller software[70] utilizing the template PDB 7LX2.

Similarly, for the BG505-SOSIP Env, PDB ID 4ZMJ was used as an initial model. These models were fitted into their respective maps using UCSF ChimeraX[83]. Coot[71] and Isolde[72] were used to fix clashes and Ramachandran outliers, as well as to add glycans. Refinement was done using real-space refinement in Phenix[73] using global minimization, NQH flips, local grid search, secondary structure restraints, Ramachandran restraints and a nonbonded weight of 1000.

For conformational flexibility and asymmetry analysis, the respective final models were rigid fitted using real space refinement in Phenix[73] into ten subclasses of unliganded 1059 and BG505 SOSIPs. These models were used to calculate the interprotomer distance analysis and root mean square deviation to compare the asymmetry and flexibility respectively in unliganded 1059 and BG505 SOSIPs. Asymmetry level of each subclass was calculated using the following equation:

$$\text{Asymmetry}_{343} \text{ level} = |Prot_{AB} - GeoM| + |Prot_{BC} - GeoM| + |Prot_{CA} - GeoM|$$

$Prot_{xy}$ = the distance between residue 343 of protomer X and residue 343 of protomer Y in each subclass.

$GeoM$ = geometric mean of 3 distances between residue 343 of all 3 protomers (AB, BC, and CA) in the final structure reconstructed from all particles for each related strain ($GeoM_{1059\text{-}SOSIP}$ = 92.47 Å; $GeoM_{BG505\text{-}SOSIP}$ = 93.17 Å). All calculations are provided in Supplementary Table 4.

### Motion analysis

The density motion analysis of unliganded 1059 and BG505 SOSIPs was performed using 3D Variability analysis as previously described[74]. The analysis used the consensus pose of the complete unliganded 1059-SOSIP cryo-EM dataset containing 737,588 particles. Similarly, in case of BG505-SOSIP, the analysis used the consensus pose of the complete unliganded BG505 SOSIP cryo-EM dataset containing 975,399 particles. Three principal component variability modes were calculated for each dataset using the default options and the low-pass filter resolution was set to 5 Å. The movies are shown in the Supplementary movies 1 and 2.

### Structural analyses

Structure figures and movies were created using PyMOL (Schrödinger) and UCSF ChimeraX[75]. Structure models were fit in maps by rigid body fit.

### Reporting summary

Further information on research design is available in the Nature Portfolio Reporting Summary linked to this article.

## Data availability

Data are available in the manuscript and supplementary files and movies. The cryo-EM maps have been deposited in the Electron Microscopy Data Bank (EMDB) under the following accession codes: EMD-41246 for unliganded 1059 SOSIP, EMD-41244 for unliganded BG505 SOSIP. The refined coordinates have been deposited in the RCSB database under the following accession codes: PDB ID 8TGW for unliganded 1059-SOSIP and PDB ID 8TGU for unliganded BG505-SOSIP. Source data are provided with this paper.

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

## Acknowledgements

We thank B.H. Hahn (University of Pennsylvania), G.M. Shaw (University of Pennsylvania), and K. Wagh (Los Alamos National Laboratory) for reading the manuscript and providing critical comments. We thank the NIH AIDS/HIV Reagent Program, Division of AIDS, NIAID, NIH for providing the following anti-HIV-1 Env antibodies: VRC01, VRC03, 3BNC117, N6, NIH45-46 G54W, F105, PG9, PG16, PGT145, 10-1074, PGT121, 10E8, 7H6, 4E10, and 35O22; the psPAX2 plasmid; HIV-1BaL; HIV-1 Subtype B Panel of SGA gp160 Env Clones (from B.H. Hahn, B.F. Keele and G.M. Shaw) and T20. We also thank D. Easterhoff, T. Bradley, and B. Haynes (Duke University) for providing the 902090 expression plasmids; J. Robinson (Tulane University) for the 17b, 19b, and JR52 expression plasmids; J. Mascola (NIH Vaccine research Center) for the VRC34 expression plasmids; R.W. Sanders and I. Del Moral Sanchez (University of Amsterdam) for providing the BG505 SOSIPv6 expressing plasmid; J. Hoxie (University of Pennsylvania) for the SupT1.CCR5 cell line; S. Ahmed and T. Picard (University of Minnesota) for helping to purify gp120; and G. Hart (University of Minnesota) for help with the flow cytometry. We thank L. Corey, J. Hural, N. Na (Fred Hutchinson Cancer Center), D. Montefiori (Duke University), those that sequenced the viral genomes, generated the expression plasmids and conducted the viral neutralization assays as well as the volunteers in the HVTN 704 trial for the Env-expressing plasmids from the AMP trial (HVTN 704). Cryo-EM data for the lectin-purified unliganded BG505-SOSIP and 1059-SOSIP were collected at the Duke Krios at the Duke University Shared Materials Instrumentation Facility (SMIF), a member of the North Carolina Research Triangle Nanotechnology Network (RTNN), which is supported by the National Science Foundation (award number ECCS-2025064) as part of the National Nanotechnology Coordinated Infrastructure (NNCI). A.H. is the recipient of an amfAR Mathilde Krim Fellowship in Basic Biomedical Research (108501-53-RKNT) and a phase II amfAR research grant (109285-58-RKVA) for independent investigators. This work was supported by an AIRP grant from the University of Minnesota Medical School (to A.H.; work on 1059-SOSIP), Avenir Award 1DP2DA049279-01 (NIH Director's New Innovator Award) from NIH/NIDA (to A.H.; work on HIV-1 evolution in vivo), National Institute of Allergy and Infectious Diseases (NIAID) U01 grant 1U01AI169587 (to A.H. (contact PI) and P.A. (work on 1059-SOSIP structure)), and NIAID R01 1R01AI167653 (to A.H.; work on N6 antiviral activity). The work was also supported by the NIH R01 AI145687 (to P.A.) and NIH U54 AI170752 (to P.A.). The content is solely the responsibility of the authors and does not necessarily represent the official views of the NIH.

## Author contributions

A.H. designed the experiments; S.R., M.H., H.C., D.P., and A.H. performed the experiments. K.R.P., R.P., X.H., S.S., K.J., and P.A. performed the structural studies and analysis of unliganded 1059 and BG505 SOSIPs. S.R., H.C., J.S., P.A., and A.H. analyzed the data; A.H. wrote the paper with contributions from all authors.

## Competing interests

A.H. is an inventor on a provisional patent application filed by the University of Minnesota for engineering 1059 SOSIP immunogens and the founder of SyntIV LLC. Other authors declare no competing interests.
