## [Peer Review File · Nature Communications]

Conformational flexibility of HIV-1 envelope glycoproteins modulates transmitted / founder sensitivity to broadly neutralizing antibodiesEditorial Note: This manuscript has been previously reviewed at another journal. This document only contains reviewer comments and rebuttal letters for versions considered at Nature Communications.

REVIEWER COMMENTS

Reviewer #1 (Remarks to the Author):

I believe that the authors have addressed my previous concerns in the last version of the revised manuscript. Removing the DEER data, which have their limitations and are complicated to interpret, should not affect the main conclusion supported by other data. SOSIP constructs often produce some misfolded trimers or other oligomeric forms that are related to protein folding not purification. Galanthus nivalis lectin (GNL) chromatography is generally considered a mild purification method that would not alter sample homogeneity in any significant way when 1059 SOSIP is eluted in 1M methyl- α -D-mannopyranoside, which is presumably properly buffered at a certain salt concentration. The protein is further purified by size-exclusion chromatography remove misfolded SOSIP species. Cryo-EM selects a subset of particles for high-resolution structure determination in any case and any impact of structural heterogeneity introduced by the Galanthus nivalis lectin column would be minimal.

Reviewer #2 (Remarks to the Author):

Parthasarathym et al. demonstrate that conformational flexibility of the HIV-1 envelope trimer modulates transmitted founder virus sensitivity to broadly neutralizing antibodies using multiple independent lines of evidence with characterization of 13 transmitted/founder strains finding 6 to be incompletely closed, with structure of partially closed envelope trimer from strain 1059, and with reconstruction of the evolutionary pathway of the CH040 envelope. They also demonstrate that the very broad antibody N6 is able to overcome this conformational flexibility, explaining in part its exceptional breadth.

Overall, this paper is well written and figures nicely presented - providing insights for the field into how conformational masking of the HIV-1 envelope can allow for resistance to neutralization.

I reviewed a prior version of this paper positively but was asked whether the DEER data that was in the original manuscript was critical or if the fact that Galanthus nivalis lectin columns used to prepare the 1059 SOSIP samples can yield structurally heterogeneous samples is an issue. I feel the DEER data was confirmatory, but not crucial to the manuscript, and feel that the Galanthus nivalis lectin columns do not introduce heterogeneity, so that the heterogeneity of the purified sample reflects the actual heterogeneity of the envelope trimer; the fact that the homogeneous closed BG505 SOSIP trimers were purified on Galanthus nivalis lectin columns, serves as a control for use of this matrix.